# Procrastination, Perfectionism, Narcissistic Vulnerability, and Psychological Well-Being in Young Adults: An Italian Study

**DOI:** 10.3390/ijerph21081056

**Published:** 2024-08-12

**Authors:** Massimiliano Sommantico, Jacopo Postiglione, Elisabetta Fenizia, Santa Parrello

**Affiliations:** Department of Humanities, University of Naples Federico II, Via Porta di Massa 1, 80133 Naples, Italy; jacopo.postiglione@gmail.com (J.P.); elisabetta.fenizia@unina.it (E.F.); parrello@unina.it (S.P.)

**Keywords:** procrastination, perfectionism, narcissism, parental expectations and criticism, young adults

## Abstract

Procrastination is generally regarded as a dysfunctional tendency to postpone tasks, due to its consequences on performance and psychological well-being. Previous research has indicated that it is linked to perfectionism and narcissism, but with mixed results. The present study explored the interaction between procrastination, perceived parental expectations, multidimensional perfectionism, and narcissism in a sample of 548 Italian young adults aged 18–35 years (*M* = 23.9; *SD* = 4.3). Participants completed an online survey consisting of a sociodemographic questionnaire and psychometric measures assessing the constructs of interest. The results showed that: (a) procrastination was positively correlated with socially prescribed perfectionism only, which, in turn, was positively correlated with perceived parental expectations and criticism, and both narcissistic grandiosity and vulnerability; (b) perceived parental expectations and criticism and narcissistic vulnerability had a positive effect on socially prescribed perfectionism, while procrastination had a negative one; and (c) narcissistic vulnerability mediated the relationship between socially prescribed perfectionism and procrastination. Taken together, the findings contribute to a better understanding of the link between procrastination, perfectionism, and narcissism in young adults, and highlight the relevance of contemporary parenting styles and the current sociocultural background for understanding dilatory behaviors.

## 1. Introduction

Procrastination can be described as the tendency to postpone relevant and prioritized activities, replacing them with pleasant activities or tasks of less importance or urgency [1]. Steel [2], who provides the most widely accepted definition in the literature, describes it as the voluntary delay of a planned course of action, despite expecting to be worse off for the delay, emphasizing the voluntary and irrational nature of the tendency to procrastinate. Research on procrastination has shown that procrastination behavior can relate to specific areas of life, such as academia (i.e., academic procrastination), or take the form of a more pervasive and chronic tendency [3]. In both cases, the negative nature of the conduct has been highlighted [4,5], which is found to be associated not only with lower performance [6], but also with various indicators of psychological distress. Indeed, different studies have shown how procrastination correlates positively with anxiety and depression [3,7,8], stress [5,9,10], feelings of shame [11,12,13], guilt [14,15], low self-esteem [16,17,18], and lower satisfaction with life [19,20]. Because of its negative influence on people’s autonomy [21,22], research has mainly focused on the identification of possible causes of procrastination. Findings have shown that it is not just a manifestation of laziness or the product of inadequate time management strategies, but the result of a complex process involving behavioral, cognitive, and affective components [11,23,24]. As Yan and Zhang’s bibliometric analysis points out [25], since the early days of research on procrastination, researchers have been interested in the study of its antecedents, among which factors related to task characteristics, such as task aversiveness, understood as the degree to which the task results are unpleasant for an individual [24], and especially dispositional characteristics, including perfectionism.

Perfectionism is described as a personality trait characterized by the pursuit of impeccability and the imposition of extremely high standards, with a tendency to make overly critical assessments about one’s behavior [26,27]. As highlighted by Damian and colleagues [28], there are two main dominant models in the literature. Each model identifies different dimensions of the construct, distinguishing between personal and interpersonal components. Hewitt and Flett [27] identified three dimensions: self-oriented perfectionism, which is the tendency to expect perfection from oneself and to overexert oneself to achieve unrealistic standards; other-oriented perfectionism, which is the tendency to expect high performance from significant others and to negatively evaluate them when the imposed standards are not achieved; and socially prescribed perfectionism, which is the tendency to believe that people have high expectations of oneself and to believe that they are subject to negative evaluations if they are not met. On the other hand, Frost and colleagues [26], identified six dimensions of perfectionism: a tendency to expect high-level performance from oneself (personal standards), excessive concern about making mistakes (concern over mistakes), excessive concern about the quality of one’s actions (doubting of actions), the perception of high parental expectations (parental expectations), the perception of hypercritical and punitive parents in case of failure (parental criticism), and a tendency to place high value on order and organization (organization). Comparing the two models, two superordinate aspects of perfectionism were recognized: (a) perfectionistic strivings (self-oriented perfectionism and personal standards), traditionally conceived as adaptive, representing the self-focused perfectionistic tendencies characterized by the pursuit of perfection, extremely high personal performance standards, and lower negative self-evaluation, and (b) perfectionistic concerns (socially prescribed perfectionism, concern over mistakes, doubting of actions), traditionally conceived as maladaptive, which refers to the concern about making mistakes due to a fear of negative social judgment in case of failure and higher negative self-evaluation [29,30].

Several studies have highlighted the influences that family may have on perfectionistic tendencies [31]. In particular, some authors [32,33] have pointed out that the dimensions of parental expectations and criticism may not be aspects of perfectionism, but factors that can promote its development. In this regard, Enns and colleagues [34] highlighted that harsh parenting—a parenting style characterized by a lack of care and high criticism, high expectations, and parental control—predicted maladaptive perfectionism, but not adaptive perfectionism. Furthermore, McArdle and Duda [35] empirically verified how the dimensions of concern over mistakes and doubts about actions were positively predicted by perceived parental expectations and criticism, while the dimension of personal standards was only predicted by perceived high parental expectations. Regarding the specific dimensions identified by Hewitt and Flett [27], in a longitudinal study of 381 adolescents aged 15–19 years, Damian and colleagues [28] found that perceived high parental expectations predicted increased levels of socially prescribed perfectionism at 7-9 months.

Regarding the association between perfectionism and procrastination, it has been widely shown how perfectionism predicts the tendency to procrastinate, but in different ways depending on personal and interpersonal dimensions. People with higher levels of self-oriented perfectionism seem to procrastinate less, while people with higher levels of socially prescribed perfectionism seem to procrastinate more [36,37]. This appears consistent with studies showing that perfectionistic strivings are negatively associated with procrastination, while perfectionistic concerns positively correlate with the tendency to procrastinate [38,39,40]. Moreover, empirical evidence suggests that perceived high parental expectations and criticism may be predictors of the tendency to procrastinate in academic settings [29,41,42], according to those studies highlighting the greater negative effects on psychological well-being of the social aspects associated with perfectionism [30,43]. Less attention has been paid to the link between procrastination and other-oriented perfectionism, which is not considered an aspect of either perfectionistic strivings or perfectionistic concerns, but a distinct dimension related to the Dark Triad [44], a configuration of personality traits composed of Machiavellianism, psychopathy, and narcissism.

Several theoretical models emphasize perfectionism as one of the central aspects of narcissism [45,46], and the pursuit of perfection is understood as a necessary strategy for the individual to maintain a vacillating and fragile sense of self and protect their self-esteem [47,48]. This conceptualization is in line with the definition of narcissism as the ability to maintain a positive self-image through a variety of affective and interpersonal self-regulatory processes [49]. Furthermore, several authors have proposed a dimensional perspective of narcissism, emphasizing the existence of a continuum from the healthy side, characterized by self-confidence and high self-esteem, to the pathological and psychotic side, characterized by marked exploitation and interpersonal manipulation [45,50,51]. Alongside a quantitative distinction, there is agreement that narcissism has two different phenotypic expressions, referred to as narcissistic grandiosity and narcissistic vulnerability [49,52,53], which may manifest more or less explicitly [54]. Narcissistic grandiosity, which captures the prototypical description of narcissism, is characterized by exaggerated exaltation of one’s image and abilities, exhibitionism, conceit, and a tendency to exploitation, while narcissistic vulnerability is characterized by an external need for recognition and a sense of one’s worth strongly anchored to that recognition [55]. Empirical studies that have analyzed the relationship between the two components of narcissism and the three dimensions of perfectionism, identified by Hewitt and Flett [27], have shown mixed results. Indeed, Flett and colleagues [56] tested for this association in two samples of 229 and 168 subjects, finding in both samples a positive association of self-oriented and socially prescribed perfectionism with narcissistic grandiosity and vulnerability, while in only one of the two samples, other-oriented perfectionism was found to be positively associated exclusively with narcissistic grandiosity. Other studies agree that other-oriented perfectionism is positively associated with narcissistic grandiosity, while socially prescribed perfectionism with narcissistic vulnerability [55,57]. This evidence is not surprising, since in the latter there is a greater focus and concern about external evaluations, understood as a tool for managing one’s self-esteem [58]. Indeed, a meta-analysis conducted by Smith and colleagues [57] confirms this finding, but also shows a positive association between self-oriented perfectionism and narcissistic grandiosity, once the overlap effect with other-oriented perfectionism is controlled for.

Although narcissism is conceptually and empirically associated with perfectionism, few studies have been conducted on the link between this construct and the tendency to procrastinate. Burka and Yuen [59] identified procrastination as a way to respond to wavering self-esteem. Similarly, Lyons and Rice [60] suggested that avoidance procrastination, a form of chronic procrastination due to a fear about one’s abilities and insecurity about performance [61], may represent a way to cope with low self-esteem, a common feature of both narcissism and procrastination [16]. In addition, Meng and colleagues [62] argue that it is likely that people with Dark Triad traits are more likely to procrastinate to cope with the pressure and competition typical in certain fields, such as academia. Research investigating the association between narcissism and procrastination has shown incongruous results, with positive [60,63] and negative correlations [64]. To our knowledge, in studies on the relationship between procrastination and narcissism, no rating scales have been used to distinguish vulnerability from grandiosity. Instead, the distinction between the two dimensions of narcissism seems to be relevant because of the different correlations they have with indicators of psychological well-being. For example, if narcissistic grandiosity turns out to be associated with externalizing disorders, higher self-esteem, and lower negative emotions, narcissistic vulnerability turns out to be associated with internalizing disorders (i.e., depression and anxiety), lower self-esteem, greater difficulty in emotion regulation, and negative affectivity [53,65,66,67]. So, those with higher levels of narcissistic grandiosity appear to be more extroverted, enthusiastic, and likely to be committed to achieving their goals [68,69], while those with higher levels of narcissistic vulnerability tend to be introverted, inhibited, and hypersensitive to criticism and failure [70,71]. These characteristics can affect the way individuals cope with daily tasks and approach performance. Indeed, Foster and Trimm [72] showed that narcissistic grandiosity positively predicts approach motivation and negatively predicts avoidance motivation, while narcissistic vulnerability only and positively predicts avoidance motivation. Finally, Manley and colleagues [73] explored the relationship between the two phenotypic expressions of narcissism and goal persistence, showing the positive effect of narcissistic grandiosity and the negative effect of narcissistic vulnerability.

Indeed, in this area young adults represent an interesting target audience because, having emerged from the adolescent psycho-social moratorium, they are in a long phase of biographical transition [74] that requires the ability to choose, commit, and manage their time according to a projected personal and professional future, while managing the emotions associated with inevitable frustrations, slowdowns, and difficulties [75,76]. The contemporary family [77], in which they are raised, has in recent decades turned out to be a context characterized by the narcissistic fragility of parents, who tend to overparent [78] and often have high expectations of success from their children, thus hoping to feel gratified in their role [79]. Contemporary society, for its part, is characterized by extreme competitiveness and pushes for the rapid achievement of success, at the expense of building bonds and valuing mistakes [80]. Within this framework, it is not strange that young adults turn out to be a vulnerable population in many studies, as during the COVID-19 pandemic [81,82,83], which has affected their time perspective, increased intolerance to uncertainty and caused a negative vision of the future [84,85].

In the Italian context, the transition to adulthood appears to be particularly slow and problematic [86,87,88], due to both the prevalence of parental styles characterized by hyperprotection, psychological control, and the discouragement of children’s autonomy [89,90,91,92], as well as socioeconomic conditions, including a lack of housing and high rates of youth unemployment, especially in the south [93]. In fact, ISTAT [93] extends the transition phase to adulthood up to the age of 35. The interaction of these specific cultural factors may influence the link between procrastination, perfectionism, and narcissism in young adults.

### Aim and Hypotheses

Based on the above, the main objective of this study was to investigate the presence of significant associations between perceived parental expectations and criticism, different phenotypic expressions of narcissism, multidimensional perfectionism, and procrastination in a sample of young Italian adults (aged 18–35).

The literature has specifically explored the academic procrastination of college-educated young adults [94], while several gaps persist on the link between some of the variables considered in this study. In particular, there is a lack of evidence on: (a) the link between perceived parental expectations with the two expressions of narcissism; (b) the association between narcissistic vulnerability and procrastination; (c) the role of procrastination as a factor that can affect the interpersonal dimension of perfectionism; and (d) the link between perceived parental expectations and nonacademic procrastination.

To address these gaps and based on research showing that narcissistic vulnerability, procrastination, and socially prescribed perfectionism share several correlates, including low self-esteem, concerns about others’ judgment, experiences of anxiety, and avoidance tendencies [43,95,96], we found it useful to consider these constructs together and we hypothesized that:

**H1a.** 
*Procrastination has a significant positive relationship with socially prescribed perfectionism and a significant negative relationship with self-oriented perfectionism.*


**H1b.** 
*Procrastination has a significant positive relationship with high parental expectations and criticism.*


**H1c.** 
*Procrastination has a significant positive relationship with narcissistic vulnerability and a significant negative relationship with narcissistic grandiosity.*


**H2a.** 
*Socially prescribed perfectionism has a significant positive relationship with narcissistic vulnerability and a significant negative relationship with narcissistic grandiosity.*


**H2b.** 
*Self-oriented and other-oriented perfectionism have a significant positive relationship with narcissistic grandiosity and a significant negative relationship with narcissistic vulnerability.*


**H3.** 
*Procrastination, narcissistic vulnerability, and high parental expectations and criticism predict socially prescribed perfectionism.*


**H4.** 
*Narcissistic vulnerability mediates the relationship between socially prescribed perfectionism and procrastination.*


## 2. Methods

### 2.1. Participants and Procedure

Participants were recruited in Italy via the Internet, through social media ads, and the informal network of acquaintances, between 4 March 2023 and 31 May 2023. The following inclusion criteria were selected: at least 18 years of age and a maximum of 35 years of age. In addition, snowball sampling was adopted, whereby the recruited participants were asked to identify other potential subjects through their social networks, and so on. Participation in the study was voluntary, anonymous, and unpaid. All subjects gave their consent to participate before proceeding to fill out the survey. The informed consent included detailed information about the objectives and procedures of the study, and the confidentiality and anonymity of the responses. Data collection measures, administered through an online survey on the Qualtrics Experience Management (XM) platform, were used to assess the variables of interest. After responding to a sociodemographic questionnaire, in which they were asked to provide some personal information (i.e., age, biological sex, gender identity, sexual orientation, educational level, employment status, housing conditions, and region of residence), participants completed the following questionnaires: the Adult Inventory of Procrastination (AIP) [97], the Parental Expectations and Criticism (PEPC) scale [26], the Multidimensional Perfectionism Scale (MPS) [98], and the Pathological Narcissism Inventory (PNI) [49].

A total of 548 subjects (*F* = 60.6%, *M* = 39.4%), aged 18–35 years (*M* = 23.9, *SD* = 4.3), responded to the survey. Most of the responses were from students (66.7%), residents of the Campania region (74%), and residents with a family of origin (75%).

### 2.2. Instruments

#### 2.2.1. Sociodemographic Questionnaire

The sociodemographic questionnaire asked the participants to provide information regarding their age, biological sex, gender identity, sexual orientation, educational level, employment status, housing conditions, and region of residence.

#### 2.2.2. Adult Inventory of Procrastination (AIP)

The AIP [97] is a 15-item self-report instrument that measures procrastination in different domains. The authors point out that the scale focuses on avoidance procrastination, which reflects the tendency to procrastinate to avoid failure and protect one’s self-esteem. Participants are asked to express their level of agreement or disagreement using a 5-point Likert-type scale (1 = strongly disagree; 5 = strongly agree). Examples of items are “I find that it takes me longer to do things than I would like” and “I am not very good at meeting deadlines”. The Italian version of the AIP [99] shows good internal consistency, with Cronbach’s *α* = 0.75. In the current study, Cronbach’s *α* was 0.85.

#### 2.2.3. Parental Expectations and Criticism (PEPC) Scale

The PEPC scale [26] is an 8-item self-report instrument that measures perceived parental expectations and criticism. It consists of two scales from the Frost Multidimensional Perfectionism Scale (FMPS) [26], unified in the Italian version and used in previous studies [28,35]. Participants are asked to express their level of agreement or disagreement using a 5-point Likert-type scale (1 = strongly disagree; 5 = strongly agree). Examples of items are, “In my family, only brilliant achievements are considered acceptable” and “My parents have always had higher expectations than mine for my future”. The Italian version of the PEPC scale [100] shows good internal consistency, with Cronbach’s *α* = 0.81. In the current study, Cronbach’s *α* was 0.87.

#### 2.2.4. Multidimensional Perfectionism Scale (MPS)

The MPS [98] is a 45-item self-report instrument that measures three dimensions of perfectionism: self-oriented, defined by excessive efforts aimed at achieving unrealistic standards, with strong self-criticism when goals are not met; other-oriented, defined by the expectation that significant others will perform highly; and socially prescribed, defined by the perception that other people have high expectations of them and the belief that they are subject to negative evaluations. Participants are asked to express their level of agreement or disagreement using a 7-point Likert-type scale (1 = *disagree*; 7 = *agree*). Examples of items are “When I am working on something, I cannot relax until it is perfect” (self-oriented perfectionism; MPSSO), “Everything others do must be of excellent quality” (other-oriented perfectionism; MPSOO), and “I find it difficult to meet the expectations others have of me” (socially prescribed perfectionism; MPSSP). The Italian version of the MPS [101] shows good internal consistency, with Cronbach’s *α* being 0.91 for self-oriented perfectionism, 0.77 for other-oriented perfectionism, and 0.87 for socially prescribed perfectionism. In the current study, Cronbach’s *α* was 0.91 for self-oriented perfectionism, 0.78 for other-oriented perfectionism, and 0.83 for socially prescribed perfectionism.

#### 2.2.5. Pathological Narcissism Inventory (PNI)

The PNI [49] is a 52-item self-report instrument that measures total narcissism and the two components of narcissistic grandiosity, defined by exploitation, grandiose fantasy, and self-sacrificing self-enhancement, and narcissistic vulnerability, defined by contingent self-esteem, hidden self, devaluing, and entitlement rage. Participants are asked to express their level of agreement or disagreement using a 6-point Likert-type scale (0 = *not at all like me*; 5 = *very much like me*). Examples of items are “I often fantasize that I have enormous influence on the world around me” (narcissistic grandiosity; NG) and “Sometimes I avoid people because I fear they will disappoint me” (narcissistic vulnerability; NV). The Italian version of the PNI [102] shows good internal consistency, with Cronbach’s *α* being 0.93 for pathological narcissism, 0.94 for narcissistic grandiosity, and 0.89 for narcissistic vulnerability. In the current study, Cronbach’s *α* was 0.94 for pathological narcissism, 0.86 for narcissistic grandiosity, and 0.94 for narcissistic vulnerability.

### 2.3. Statistical Analyses

Data analyses were conducted using the SPSS 29.0.1.0 package (IBM Corp, Armonk, USA) [103]. Correlation analyses were conducted by referring to Pearson’s *r* coefficient (between 0.10 and 0.29 = weak association; between 0.30 and 0.49 = moderate association; >0.50 = strong association; *p* < 0.05) [104]. Group differences were tested by the independent *t*-test and ANOVA test (*p* < 0.05) and the effect sizes were measured using Cohen’s *d* (*d* ≥ 0.2 = small; ≥0.5 = medium; ≥0.8 = large) and the eta squared indexes (*η*^2^ ≥ 0.01 = weak; ≥0.059 = moderate; ≥0.138 = strong) [104]. Multiple linear regression analyses were conducted using standardized *β* coefficients and *R*^2^ coefficients (*p* < 0.05). In order to investigate the process through which one variable influences another via an intermediary variable, we conducted mediation analysis that enables a more in-depth examination of the relationship between variables. Through the PROCESS macro tool for SPSS [105], direct and indirect effects were examined using the bootstrapping method to estimate the confidence intervals (CIs), adjusted by resampling with 5000 resamples. Confidence intervals that do not include zero indicate a significant effect [106].

## 3. Results

### 3.1. Descriptive Statistics and Group Differences

The mean and standard deviation of the study variables, *t*-test results examining gender differences, and Cronbach’s *α* are shown in Table 1. The mean for the AIP was 38.5 (*SD* = 10.6); the mean for the PEPC scale was 20.3 (*SD* = 7.3); the mean for MPSSO, MPSOO, and MPSSP was, respectively, 70.1 (*SD* = 17.7), 51.8 (*SD* = 12.4), and 53.5 (*SD* = 13.9); and the mean for the PNI varied between 3.7 for narcissistic grandiosity (*SD* = 0.7) and 3.4 for narcissistic vulnerability (*SD* = 0.8).

As seen in Table 1, the *t*-test analyses showed significant gender differences. Male participants reported significantly higher scores than female participants on the scales of other-oriented perfectionism (*M_F_* = 50.7 vs. *M_M_* = 53.3; *t*(_546_) = 2.40, *p* < 0.05; *d* = 0.21) and narcissistic grandiosity (*M_F_* = 3.6 vs. *M_M_* = 3.8; *t*(_546_) = 2.74, *p* < 0.05; *d* = 0.24), while female participants scored significantly higher than males on the scales of self-oriented perfectionism (*M_F_* = 71.7 vs. *M_M_* = 67.3; *t*(_546_) = −2.86, *p* < 0.005; *d* = 0.25), socially prescribed perfectionism (*M_F_* = 54.5 vs. *M_M_* = 52.1; *t*(_509,559_) = −2.10, *p* < 0.05; *d* = 0.17), and narcissistic vulnerability (*M_F_* = 3.4 vs. *M_M_* = 3.2; *t*(_546_) = −2.92, *p* < 0.005; *d* = 0.25).

The ANOVA showed significant group differences regarding employment. Compared with the unemployed and working groups, students reported significantly higher levels of procrastination (*M_I_* = 39.4 vs. *M_II_* = 38.3 vs. *M_III_* = 36.5; *F*(_2,545_) = 3.835, *p* < 0.05; *η*^2^ = 0.01), total narcissism (*M_I_* = 3.5 vs. *M_II_* = 3.4 vs. *M_III_* = 3.3; *F*(_2,545_) = 5.403, *p* < 0.01; *η*^2^ = 0.01), narcissistic grandiosity (*M_I_* = 3.8 vs. *M_II_* = 3.5 vs. *M_III_* = 3.6; *F*(_2,545_) = 4.087, *p* < 0.05; *η*^2^ = 0.01), and narcissistic vulnerability (*M_I_* = 3.4 vs. *M_II_* = 3.3 vs. *M_III_* = 3.2; *F*(_2,545_) = 4.973, *p* < 0.01; *η*^2^ = 0.02).

No other significant group differences emerged.

### 3.2. Correlations

The correlations between participants’ age and the study variables are presented in Table 2. The results indicate that procrastination had a statistically significant positive correlation with perceived high parental expectations and criticism (*r* = 0.24; *p* < 0.01), socially prescribed perfectionism (*r* = 0.12; *p* < 0.05), total narcissism (*r* = 0.28; *p* < 0.01), narcissistic grandiosity (*r* = 0.20; *p* < 0.01), and narcissistic vulnerability (*r* = 0.28; *p* < 0.01), and a statistically significant negative correlation with self-oriented (*r* = −0.09; *p* < 0.05) and other-oriented (*r* = −0.10; *p* < 0.05) perfectionism; perceived high parental expectations and criticism showed a statistically significant positive correlation with self-oriented (*r* = 0.19; *p* < 0.01), other-oriented (*r* = 0.10; *p* < 0.05), and socially prescribed (*r* = 0.68; *p* < 0.01) perfectionism, but also with total narcissism (*r* = 0.46; *p* < 0.01), narcissistic grandiosity (*r* = 0.29; *p* < 0.01), and narcissistic vulnerability (*r* = 0.47; *p* < 0.01); finally, the three components of perfectionism had a statistically significant positive correlation with total narcissism (MPSSO: *r* = 0.33; *p* < 0.01; MPSOO: *r* = 0.20; *p* < 0.01; MPSSP: *r* = 0.47; *p* < 0.01), narcissistic grandiosity (MPSSO: *r* = 0.27; *p* < 0.01; MPSOO: *r* = 0.25; *p* < 0.01; MPSSP: *r* = 0.25; *p* < 0.01), and narcissistic vulnerability (MPSSO: *r* = 0.32; *p* < 0.01; MPSOO: *r* = 0.15; *p* < 0.01; MPSSP: *r* = 0.51; *p* < 0.01).

### 3.3. Regression Analyses

Hierarchical multiple regression analyses were conducted to determine the extent to which some of the variables analyzed contribute to the prediction of socially prescribed perfectionism. The model that emerged is summarized in Table 3. After controlling for differences in age and gender, the addition of AIP to the prediction of socially prescribed perfectionism resulted in a statistically significant increase in *R*^2^ of 0.017 (*F*_(1,544)_ = 9.365, *p* < 0.01). The addition of NV to the socially prescribed perfectionism prediction resulted in a statistically significant increase in *R*^2^ of 0.250 (*F*_(1,543)_ = 187.168, *p* < 0.001). The addition of the PEPC scale to the socially prescribed perfectionism prediction resulted in a statistically significant increase in *R*^2^ of 0.258 (*F*_(1,542)_ = 299.247, *p* < 0.001). The full model of age, gender, AIP, NV, and the PEPC scale to predict socially prescribed perfectionism turns out to be statistically significant, *R*^2^ = 0.533 (*F*_(5,547)_ = 123.625, *p* < 0.001; *R*^2^ adjusted = 0.53).

### 3.4. Mediation Analyses

The direct and indirect effects of socially prescribed perfectionism on procrastination were explored through the variable of narcissistic vulnerability. No direct effects were found, while indirect effects were observed with a coefficient of 0.11 (IC 95% [0.07, 0.15]). Figure 1 graphically depicts the mediation model and shows the coefficients for the observed direct effects.

Details on the mediation analyses are given in Table 4.

## 4. Discussion

The ability to manage one’s time in a balanced manner is considered to be an important indicator of subjective well-being [107], while the tendency to procrastinate is usually associated with various forms of psychological distress [3,5,15,20]. In young adults, dilatory conduct can significantly hinder the developmental tasks associated with the transition to adulthood. Indeed, at this age, individuals must abandon the adolescent moratorium that allowed for an intense exploration and a suspension of commitments in the transition to adulthood that requires choices of existential, educational, and/or work paths, and thus the ability to manage their time as best they can. The pressure of family and social expectations can make this transition particularly difficult, especially in those who have developed traits of narcissism and perfectionism (self-oriented, other-oriented, and socially prescribed). In this study we were able to investigate the interplay between these dimensions, specifically hypothesizing a central role of narcissistic vulnerability, which has not been highlighted in the existing literature.

### 4.1. Correlations between Procrastination, Narcissism, and Perfectionism

Both H1a and H1b were confirmed: consistent with the literature on the link between procrastination and perfectionism [36,37], our results indicate statistically significant positive correlations of procrastination with socially prescribed perfectionism and statistically significant negative correlations with self-oriented perfectionism. These findings corroborate evidence indicating that maladaptive aspects of perfectionism are associated with the tendency to put off tasks [38,39,40].

Our study extends what has been observed in studies on the link between academic procrastination and parental expectations and criticism [29,41,42], showing that the latter have positive correlations with the tendency to procrastinate not only in academic contexts (H1b). These results are in line with the observations by Flett and colleagues [108] that the association of high parental expectations with high parental criticism are important precursors of the tendency to procrastinate. As Mitchell [109] notes, a parenting style characterized by a combination of high criticism and high expectations on children fosters the internalization of a negative self-appraisal, characterized by shame and feelings of worthlessness. It is possible to hypothesize that procrastination is a behavior aimed at preventing parents’ negative judgments. However, this strategy may prove to be ineffective, since the fear of not being able to meet others’ expectations may induce a vicious cycle of procrastination [110].

H1c was partially supported, since procrastination was found to be positively associated with both narcissistic grandiosity and narcissistic vulnerability. As for the association between procrastination and narcissistic vulnerability, to our knowledge, this link has never been studied previously. Some of the main characteristics of narcissistic vulnerability are intense shame about needs and ambitions, low self-esteem, and hypersensitivity to threats [49,54]. In this vein, our results appear to be consistent with contributions in the literature on the positive association of procrastination with low self-esteem [2,16,17,18] and shame [11,12,13,15], the latter central to the experiences of Italian adolescents and young adults who are constantly exposed to unrealizable models of success and social and parental pressures [79,111]. Moreover, the results are in line with studies emphasizing the avoidance tendency of people with a greater sense of narcissistic vulnerability [72,112], who are inclined to defend themselves against possible threats to their value, such as failure in a task. In contrast, the data on the association between narcissistic grandiosity and procrastination appear at odds with the literature, which shows that people who have higher levels of narcissistic grandiosity should be willing to be committed and persistent in achieving their goals [68,69,73]. We can hypothesize that people with greater narcissistic grandiosity might consider themselves able to defy time, so they should procrastinate in the belief that they can accomplish tasks. In this vein, procrastination might represent one of the self-enhancement strategies typical of narcissistic grandiosity [54].

### 4.2. Correlations between Perfectionism and Narcissism

Furthermore, both hypotheses regarding the different dimensions of perfectionism (H2a and H2b) were partially confirmed. In fact, both grandiosity and narcissistic vulnerability showed positive statistically significant correlations with all dimensions of perfectionism. These results could be related to the fact that we did not control for the overlap effect between the different dimensions of perfectionism, as carried out in other studies that showed some differences in the association between the components of narcissism and perfectionism [55,57]. Despite this, and consistent with observations that people who score higher for narcissistic vulnerability are more concerned with external evaluations [58], in this study the only strong association was found between narcissistic vulnerability and socially prescribed perfectionism.

### 4.3. The Predictive Role of Procrastination, Perfectionism, and Narcissism on Socially Prescribed Perfectionism

Multiple regression analyses, conducted to investigate the contribution of procrastination, perceived high parental expectations and criticism, and narcissistic vulnerability on the explanation of socially prescribed perfectionism, indicated a positive effect of perceived high parental expectations and criticism and narcissistic vulnerability and a negative effect of procrastination on socially prescribed perfectionism (H3). As highlighted by Burka and Yuen [113], since procrastinators observe the negative consequences of procrastination on their performance, they might deny or downplay their perfectionistic tendencies when responding to self-report instruments that ask them to identify or not identify themselves as perfectionists. This observation is also consistent with the weak correlations between the dimensions of perfectionism and procrastination found in this study and elsewhere [2,108].

Finally, H4 was confirmed: mediation analyses suggest that socially prescribed perfectionism does not directly affect procrastination, in agreement with what Sherry and colleagues [114] observed, but exerts an indirect effect through narcissistic vulnerability. The influence of narcissistic vulnerability on procrastination, for the first time explored in this study, could be related to the function of procrastination as a defensive strategy to avoid negative emotions [115]. The hypersensitivity to rejection and humiliation and concern about external judgment that individuals characterized by greater narcissistic vulnerability develop from the experience of unbalanced affective relationships with parental figures [78,116,117,118], make them more prone to thoughts about the consequences of failing to be perfect [56,57].

Therefore, it is possible to hypothesize that narcissistic vulnerability is a particularly relevant trait to explain the mechanism, illustrated by some authors [94,113,119], whereby those who aim for perfection may procrastinate until they are certain that they will meet their and others’ expectations, thus keeping undamaged a fragile self-image. Socially prescribed perfectionism would manifest when individuals feel they must achieve high standards to meet others’ expectations. In Italian society, particularly in educational institutions, the strong emphasis on success can amplify social pressures, leading students to believe that only the achievement of excellence can enable them to gain approval and recognition. In this vein, procrastination may appear to be a valid strategy in the immediate term, but, in the long run, it can lead to catastrophic consequences, as evidenced by the dramatic suicides of young adults in universities, often associated with the stress of academic commitments and exam-related procrastination [120]. This framework adds useful information for organizing strategies to prevent and treat the mental health of college-educated young adults. University policies also incentivize competition because of a lack of economic resources: they reward high performance achieved in a short period of time, excluding access to higher education for those with lower performance levels. In addition, they do not provide adequate forms of compensatory support (educational and organizational) to those who are at a disadvantage (working students, students with cultural differences, psychologically fragile students). On the other hand, prolonged cohabitation with family, experienced by most students, does not help them to keep a distance from an educational climate defined by high expectations of success. Thus, the academic path becomes risky for mental health [121]. Unfortunately, Italian university counseling models show serious shortcomings [122,123,124]: one wonders whether they should be rethought based on the new profiles of university students and flanked by strategies of parental involvement, to reassure them about the ‘normality’ of their children’s delays and failures.

### 4.4. Group Differences Regarding Gender and Employment

The results of the *t*-test analyses indicated significant gender differences related to narcissism and perfectionism. Specifically, consistent with the literature on the topic [125,126], male participants reported higher levels of narcissistic grandiosity, while female participants reported higher scores of narcissistic vulnerability. Compared to narcissistic grandiosity, gender differences related to narcissistic vulnerability have been less explored and research has not always shown concordant results. Indeed, a meta-analysis conducted by Grijalva and colleagues [125] showed that there were no significant gender differences in levels of narcissistic vulnerability, while other research findings align with our results [127,128]. Picking up on Chrétien and colleagues’ observation about gender differences in narcissism in adolescence [129], it is possible that, starting from this developmental period, females feel intense social pressure related to socially imposed aesthetic standards. This pressure may lead to the development of a certain vulnerability about their worth, which in turn would seem to affect social anxiety related to their body image [130]. Concerning perfectionism, although it has been assumed that there are no gender differences in perfectionistic tendencies [27], we observed in male participants higher levels of other-oriented perfectionism and, in female participants, higher levels of self-oriented and socially prescribed perfectionism. These results are in line with Sherry and colleagues’ findings [131] that men are more prone than women to express narcissism by imposing perfectionistic expectations on others.

The results of the ANOVA tests also showed that, compared to the groups of working and unemployed young adults, students reported higher levels of procrastination, according to research findings showing that the tendency to procrastinate is more present among students [132,133], thus indicating that the academic environment constitutes itself as procrastination friendly [133]. In addition, students showed higher levels of total narcissism, narcissistic grandiosity, and narcissistic vulnerability. It is possible that training and academic contexts, characterized by competitiveness and emphasis on performance, provide incentives to display narcissistic traits to gain recognition and success.

### 4.5. Strengths, Limitations, and Future Research Directions

To our knowledge, this study is the first to explore the link between narcissism, perfectionism, and procrastination and to assess the specific relationship between narcissistic grandiosity and narcissistic vulnerability and procrastination. The joint consideration of some relevant characteristics for contemporary young adults, such as the social and personal imperative to be perfect, narcissism, and the role of high parental expectations and criticism of children, returns interesting information on the link between these variables and reinforces the idea that they may interact with each other in the Italian sociocultural scenario.

The use of an instrument designed to measure nonacademic procrastination is a strength within international literature, which has prevalently focused on the study of academic procrastination [25]. Another strength is the use of instruments that allowed for the distinction between intrapersonal and interpersonal aspects of perfectionism and between the two phenotypic expressions of narcissism.

Despite this, the study has some limitations. The first limitation concerns convenience sampling, which implies specific biases, such as the volunteers’ biases related to their specific characteristics. In addition, the sample consisted mainly of females and students from the Campania region in southern Italy, which may have biased the results, limiting the generalizability of the study’s findings. Future research samples should include more young working adults and be more balanced in terms of gender. Another possible limitation of the study is the mono-method, due to the fact that there may have been an amplification of the scores in terms of the observed associations, since we assessed all the variables in the study using self-report instruments.

Future investigations could also integrate quantitative and qualitative data, such as interviews or focus groups, to deeply investigate the mechanisms associated with procrastination. Furthermore, future studies could investigate differences in the results by using other instruments to assess procrastination tendencies. In this regard, despite the AIP allowing us to collect data on young adults and to measure procrastination in different life domains, it was developed to measure avoidance procrastination, a delaying behavior aimed at avoiding making one’s difficulties related to tasks visible [61]. It would be interesting to verify whether the associations between the different variables and procrastination could be confirmed by referring to other instruments. Another possible direction for future research might be to check for the overlap effect between the two dimensions of narcissism to observe possible variations in the results. Finally, a future longitudinal research design could be planned, following the subjects in later stages of adult life. This could allow for causal inferences, which were not possible in this study given its cross-sectional nature.

## 5. Conclusions

Taken together, the results of this study show how some conceptually close variables, relevant in the experience of young Italian adults, can be associated with each other, providing a first insight into the narcissism–perfectionism–procrastination link and the maladaptive aspects that may affect dilatory behaviors and psychological well-being.

In particular, the evidence gathered in the present study indicates that narcissistic vulnerability may play a crucial role in the relationship between social expectations of perfectionism and procrastination. Our findings suggest that young people who predominantly express their narcissism through this phenotypic expression may be particularly susceptible to the pressure of parental and social expectations and more likely to procrastinate to avoid the risk of failure and subsequent shame. Apparently, the current Italian cultural context contributes to reinforce issues related to procrastination and perfectionism, as well as to promote the manifestation of narcissism. This framework highlights the need for interventions aimed at supporting the adoption of effective strategies to reduce unbridled competition and evaluative stress and to curb fears arising from the pressure of social expectations.

## Figures and Tables

**Figure 1 ijerph-21-01056-f001:**
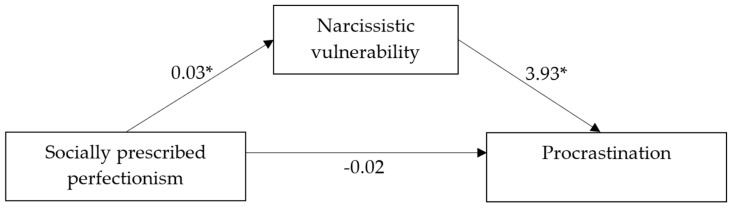
Mediation model with coefficients for direct effects. *N* = 548. * *p* < 0.001. All present effects are unstandardized.

**Table 1 ijerph-21-01056-t001:** Descriptive statistics and gender differences.

	Total Sample		Women		Men				
	(*N* = 548)		(*N* = 332)		(*N* = 216)				
	** *M* **	** *SD* **	** *M* **	** *SD* **	** *M* **	** *SD* **	** *t* **	** *d* **	** *α* **
AIP	38.5	10.6	38.4	11.2	38.7	9.6	0.31	0.02	0.85
PEPC	20.3	7.3	20.2	7.6	20.4	6.9	0.33	0.02	0.87
MPSSO	70.1	17.7	71.7	18.2	67.3	16.5	−2.86 **	0.25	0.91
MPSOO	51.8	12.4	50.7	12.6	53.3	11.9	2.40 *	0.21	0.78
MPSSP	53.5	13.9	54.5	14.7	52.1	12.5	−2.10 *	0.17	0.83
PNI	3.5	0.7	3.5	0.7	3.4	0.6	−1.19	0.10	0.94
NG	3.7	0.7	3.6	0.7	3.8	0.7	2.74 *	0.24	0.86
NV	3.4	0.8	3.4	0.8	3.2	0.7	−2.92 **	0.25	0.94

AIP = Adult Inventory of Procrastination; PEPC = Parental Expectations and Criticism scale; MPSSO = self-oriented perfectionism; MPSOO = other-oriented perfectionism; MPSSP = socially prescribed perfectionism; PNI = Pathological Narcissism Inventory (total score); NG = narcissistic grandiosity; NV = narcissistic vulnerability. * *p* < 0.05; ** *p* < 0.005.

**Table 2 ijerph-21-01056-t002:** Correlations.

	1	2	3	4	5	6	7	8	9
1. Age	-								
2. AIP	−0.077	-							
3. PEPC	−0.055	0.246 **	-						
4. MPSSO	−0.078	−0.092 *	0.199 **	-					
5. MPSOO	−0.042	−0.102 *	0.108 *	0.325 **	-				
6. MPSSP	0.029	0.126 **	0.685 **	0.377 **	0.201 **	-			
7. PNI	−0.187 **	0.285 **	0.462 **	0.339 **	0.204 **	0.470 **	-		
8. NG	−0.167 **	0.204 **	0.295 **	0.279 **	0.250 **	0.250 **	0.800 **	-	
9. NV	−0.171 **	0.285 **	0.479 **	0.321 **	0.154 **	0.510 **	0.958 **	0.594 **	-

AIP = Adult Inventory of Procrastination; PEPC = Parental Expectations and Criticism scale; MPSSO = self-oriented perfectionism; MPSOO = other-oriented perfectionism; MPSSP = socially prescribed perfectionism; PNI = Pathological Narcissism Inventory (total score); NG = narcissistic grandiosity; NV = narcissistic vulnerability. * *p* < 0.05; ** *p* < 0.01.

**Table 3 ijerph-21-01056-t003:** Hierarchical multiple regression analysis for the prediction of socially prescribed perfectionism (*N* = 548).

Predictor	*B*	*SE*	*β*	*t*	*p*	*R^2^*	*ΔR^2^*	*p*
Step 1						0.008		0.108
Age	0.315	0.097	0.097	3.248	0.001			
Gender	1.610	0.851	0.056	1.892	0.059			
Step 2						0.025	0.017	0.002
AIP	−0.112	0.041	−0.085	−2.749	0.006			
Step 3						0.275	0.250	<0.001
NV	4.589	0.609	0.264	7.539	<0.001			
Step 4						0.533	0.258	<0.001
PEPC	1.112	0.064	0.586	17.299	<0.001			

AIP = Adult Inventory of Procrastination; NV = narcissistic vulnerability; PEPC = Parental Expectations and Criticism scale.

**Table 4 ijerph-21-01056-t004:** Mediation analysis of socially prescribed perfectionism on procrastination through narcissistic vulnerability (*N* = 548).

	NV			AIP		
	*b*	*SE*	*p*	*b*	*SE*	*p*
MPSSP	0.03	0.00	<0.001	−0.02	0.03	0.58
NV	−	−	−	3.93	0.63	<0.001
Constant	1.83	0.12	<0.001	26.24	2.07	<0.001
	*R*^2^ = 0.26			*R*^2^ = 0.08		
	*F*_(1,546)_ = 192.30, *p* < 0.001			*F*_(2,545)_ = 24.19, *p* < 0.001		

AIP = Adult Inventory of Procrastination; MPSSP = socially prescribed perfectionism; NV = narcissistic vulnerability.

## Data Availability

The data presented in this study are available upon request from the corresponding author.

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
