# Peer review of "Procrastination, Perfectionism, Narcissistic Vulnerability, and Psychological Well-Being in Young Adults: An Italian Study"

_ijerph, 2024, doi:10.3390/ijerph21081056_

Round 1

Reviewer 1 Report

Comments and Suggestions for Authors

The research addresses an important issue. The theoretical background is thorough, based on an adequate number and quality of references. This is a good basis for discussion and interpretation of own results. This, however, requires a good question and hypothesis, which is not currently a strength of the study. The sample lacks a thorough developmental explanation of stages 18-35. Why was this age period chosen? It is a very broad stage, which fundamentally affects the results. The average is 24 years, so the sample is probably skewed. So who do we actually get information on? The research question is very general, it does not indicate what gap the research is trying to answer. The hypothesis thus loses all seriousness. At this age stage, and at other times, the contexts are very diverse. Serious, detailed question(s) and hypotheses should be formulated. The table presenting the regression analysis should include as much information as possible. It is perfectly understandable that a different category has been included for sex. But does it make sense to have 2 separately? I would omit this and thus examine the differences between the sexes with a t-test. Consider what you gain and what you lose by doing so. You can't claim anything about subgroup 3.  

There are a few typos and hyphenation errors in the text. I suggest re-analysing the data after some more precise research questions and hypotheses have been formulated. In this form the study does not add anything new, it is built on shaky foundations, from which it is difficult to add anything extra for theory and practice.

Comments on the Quality of English Language

See my opinion and part of Q of EL

Author Response

Reviewer 1:

  • The gaps in the literature that this study aims to fill have been made more evident in both the Aim and Hypotheses section and the Discussion. Additionally, detailed hypotheses have been formulated;
  • The section describing the developmental characteristics of young adults has been expanded. The age range of 18-35 was selected based on Italian data reported by ISTAT (National Institute of Statistics): in Italy, this age is considered a transition to adulthood (see added cultural factors in the text).
  • The table has been updated.
  • Subgroup 3 (with 2 subjects) has been eliminated, the t-test for M/F difference has been performed, and all analyses have been redone in light of the new number of subjects (from 550 to 548).
  • The data have been reanalyzed in light of the elimination of the 2 subjects from subgroup 3. The discussion of the results has been divided into sections to enhance clarity.
  • The English text has been revised.

Reviewer 2 Report

Comments and Suggestions for Authors

The manuscript "Procrastination, Perfectionism, and Narcissistic Vulnerability in Young Adults: An Italian Study" examines the relationship between procrastination, perfectionism, and narcissistic vulnerability in young adults, with a specific focus on an Italian population. This study presents significant findings that contribute to the understanding of the interplay between these psychological constructs and their impact on the mental health and behavioral tendencies of young adults. However, several areas require attention and improvement to enhance the clarity and impact of the study.

1.     The theoretical framework could be strengthened by more explicitly linking procrastination, perfectionism, and narcissism to developmental tasks and challenges in the transition to adulthood to contextualize the study's findings within the broader literature on emerging adulthood.

2.     The sample was predominantly female and consisted mainly of students from Campania, which limits the generalizability of the findings. The authors should discuss this study’s limitations and possible implications for the conclusions of the study.

3.     The cross-sectional design of the study reduced the ability to establish causality between the variables. While this limitation is mentioned, more details on how future longitudinal studies could address this issue would strengthen the discussion.

4.     The authors should provide more information on the psychometric properties of the questionnaires in the study sample.

5.     The Authors should ensure that terms and definitions are used consistently throughout the manuscript. For example, clarify the operational definitions of "narcissistic vulnerability" and "grandiosity" early in the paper and maintain these definitions consistently.

6.     I recommend a more detailed explanation of the statistical analyses, particularly mediation analysis. Although the PROCESS macro is a standard tool, a brief explanation of how it was used and interpreted in this study would be beneficial to the readers.

7.     The manuscript would benefit from a more nuanced discussion of how cultural factors specific to Italy might influence the relationships between procrastination, perfectionism, and narcissism.

8.     The Authors should include further references to studies on the well-being of young adults during the pandemic, as cited in the manuscript. Some studies recommended are: 10.3390/ejihpe13080108; 10.13129/2282-1619/mjcp-3009; 10.1371/journal.pone.0245083.

9.     Thorough proofreading is needed to correct minor grammatical errors and improve the overall readability of the manuscript.The manuscript "Procrastination, Perfectionism, and Narcissistic Vulnerability in Young Adults: An Italian Study" examines the relationship between procrastination, perfectionism, and narcissistic vulnerability in young adults, with a specific focus on an Italian population. This study presents significant findings that contribute to the understanding of the interplay between these psychological constructs and their impact on the mental health and behavioral tendencies of young adults. However, several areas require attention and improvement to enhance the clarity and impact of the study.

Comments on the Quality of English Language

Thorough proofreading is needed to correct minor grammatical errors and improve the overall readability of the manuscript.

Author Response

  • In the Introduction, the theoretical framework has been enhanced with a specific focus on young adults.
  • The limitations of the study related to sampling have been more prominently highlighted.
  • It has been concisely clarified how future longitudinal studies could address the issues present in this cross-sectional study.
  • For each questionnaire, the Cronbach's α—previously only presented in the table—has also been reported in the text.
  • The definitions of constructs related to narcissism have been standardized throughout the text.
  • A more detailed explanation of the mediation analysis has been provided.
  • In the Introduction, a section has been included on the specific cultural factors in Italy that may influence the relationships between procrastination, perfectionism, and narcissism.
  • The suggested references on the well-being of young adults during the pandemic have been included.

The English text has been revised.

Round 2

Reviewer 2 Report

Comments and Suggestions for Authors

I appreciated the revised manuscript and no further changes are needed.